# A Process-Oriented Method for Tracking Rainstorms with a Time-Series of Raster Datasets

**Cunjin Xue \*** , **Jingyi Liu, Guanghui Yang and Chengbin Wu**

Key Laboratory of Digital Earth Science, Aerospace Information Research Institute,
Chinese Academy of Sciences, Beijing 100094, China; jingyiliu24@163.com (J.L.);
yangguanghui1994@163.com (G.Y.); wcb892534877@icloud.com (C.W.)
**\*** Correspondence: xuecj@radi.ac.cn; Tel.: +86-010-82178126

**Abstract:** Extreme rainstorms have important socioeconomic consequences, but understanding their fine spatial structures and temporal evolution still remains challenging. In order to achieve this, in view of an evolutionary property of rainstorms, this paper designs a process-oriented algorithm for identifying and tracking rainstorms, named PoAIR. PoAIR uses time-series of raster datasets and consists of three steps. The first step combines an accumulated rainfall time-series and spatial connectivity to identify rainstorm objects at each time snapshot. Secondly, PoAIR adopts the geometrical features of eccentricity, rectangularity, roundness, and shape index, as well as the thematic feature of the mean rainstorm intensity, to match the same rainstorm objects in successive snapshots, and then tracks the same rainstorm objects during a rainstorm evolution sequence. In the third step, an evolutionary property of a rainstorm sequence is used to extrapolate its spatial location and geometrical features at the next time snapshot and reconstructs a rainstorm process by linking rainstorm sequences with an area-overlapping threshold. Experiments on simulated datasets demonstrate that PoAIR performs better than the Thunderstorm Identification, Tracking, Analysis and Nowcasting algorithm (TITAN) in both rainfall tracking and identifying the splitting, merging, and merging-splitting of rainstorm objects. Additionally, applications of PoAIR to Integrated Multi-satellitE Retrievals for Global Precipitation Mission (GPM/IMERG) final products covering mainland China show that PoAIR can effectively track rainstorm objects.

**Keywords:** process-oriented tracking; rainstorm object; evolution sequence; raster datasets

---

## 1. Introduction

With global warming, extreme rainstorm events tend to occur more frequently than ever [1], which may cause pluvial flooding and potentially severe socioeconomic consequences. Knowing when, where, and how rainstorm events begin and evolve is of paramount importance for science and society [2]. The development of advanced remote sensing technologies makes it possible to monitor and track these phenomena at a large scale [3,4]. However, observing the fine spatial patterns and structures and temporal evolution of rainstorms from time-series of remote sensing products still remains a challenge [5].

Generally, a rainstorm is defined as intense precipitation averaged over a time period and over a geographic region [6,7]. Characterizing rainstorms from raster datasets involves two steps:

- To identify a rainstorm object and determine its spatial location and structure at each time snapshot.
- To track the location and intensity of rainstorm objects to determine rainstorm evolution.

The general idea of the first point is to define a rainstorm object as a contiguous region of a set of connected pixels that exceed a specified threshold. The Thunderstorm Identification, Tracking,

Analysis and Nowcasting (TITAN) algorithm, a well-known object-based algorithm, used a specified single reflectivity threshold value to identify contiguous regions as rainstorm entities [8,9]. The hard single threshold is subject to one problem, that is, it is not possible to distinguish between noisy points and initiating rainstorm cells [6,10]. To address this problem, researchers have applied multiple thresholds [5,6,11,12] and combined thresholds with spatial and temporal features [7,13] to distinguish different types of rainstorm objects. Whether a single threshold or multiple ones are used, rainstorm identification is subject to seasonal, regional, and climatological variabilities [6]. To help mitigate this problem, the watershed transform algorithm has been widely used in many studies [6,10,14].

Rainstorm tracking algorithms link rainstorm objects among successive snapshots. A key factor in such algorithms is which criteria are selected to determine whether rainstorm objects are the same or not. The TITAN algorithm [8] uses a spatial overlapping threshold to determine whether a rainstorm object in snapshot T+1 is the same as the one in snapshot T. The Hungarian algorithm considers the geometrical features and moving distance among rainstorm objects in two successive snapshots. However, as it largely relies on the overlapping areas between successive rainstorm objects, TITAN has difficulty in effectively tracking fast-moving rainstorm objects [5,15]. To address these deficiencies, a number of modifications have been proposed. For example, Johnson et al. [16] replaced an overlapping area with a centroid distance to determine whether rainstorm objects belong to the same trajectory or not. Additionally, Liu et al. [12] simultaneously considered the topological overlap, centroid distance, and movement direction of rainstorm objects to design the criterion for classification as rainstorm objects. A region tree structure was first used to represent the rainstorm objects at different scales, and an area-overlapping-based method was then used to match the rainstorm objects in successive snapshots [17].

Another type of rainstorm tracking algorithm is a cross-correlation algorithm, which calculates a motion vector field and forecasts the movement of rainstorm objects [18]. For each rainstorm object, a single movement vector can be replaced by averaging the mapped movement vectors obtained from the field tracker; then, the averaged movement vector is used to extrapolate the rainstorm object in the next snapshot, which is limited by the selected matching criterion [5]. Thus, Muñoz et al. [5] adopted an object-based method to calculate a number of measurement indexes, including a distance component, an area component, a structural component, and an eccentricity component. The measured indexes were used as a cost function to quantify the similarity between rainstorm objects in successive snapshots.

Most of the above object-based and vector-based field methods used a linear regression model to infer the presence of rainstorm objects in snapshot T+1 that were present in snapshot T. However, few of these extrapolation models consider rainstorm evolution characteristics, e.g., initiation, maturity, and dissipation. To overcome this problem, Fiolleau and Roca [9] adopted a regional growth method to match rainstorm objects in successive snapshots. However, as they used the size of the spatiotemporal neighborhood as the search radius, the tracking results were sensitive to the neighborhood size, which is difficult to determine.

In spatial terms, a rainstorm is defined as a region including a group of pixels that meets a size criterion [6], while in temporal terms, it is defined as a specified duration with high values of rainfall intensity separated from other periods [19]. In reality, in spatial terms, rainstorms evolve from production through development to termination [9,17,20]. Thus, this paper designs a process-oriented algorithm for identifying and tracking a rainstorm, named PoAIR. The rest of this paper is organized as follows. Section 2 describes a semantic process of rainstorms and presents basic concepts. Section 3 designs a workflow for identifying and tracking rainstorms from a time-series of raster datasets, which includes the instantaneous retrieval of rainstorm objects, the linkage of rainstorm sequences, and the reconstruction of rainstorm processes. In Section 4, a simulated rainstorm process and a real precipitation dataset are used to evaluate our proposed algorithm. Section 5 summarizes the conclusions.

## 2. Process Semantics of Rainstorms

According to geographical process semantics [21–23], a rainstorm process consists of one or more evolution sequences, each of which includes one or more instantaneous objects. The related concepts are listed as follows, shown in Figure 1.

- A rainstorm process: A rainstorm covers a specified spatial region and evolves from production through development to termination.
- A rainstorm sequence: A successive series of rainstorm objects which have similar spatial structures and thematic features.
- An instantaneous rainstorm object: A rainstorm region which has a clearly defined spatial range in a given snapshot (i.e., at a given time).
- A rainstorm region: Rainstorm cells which connect with each other in spatial neighborhoods.
- A rainstorm cell: A grid cell in a raster dataset which exceeds a rainfall threshold.

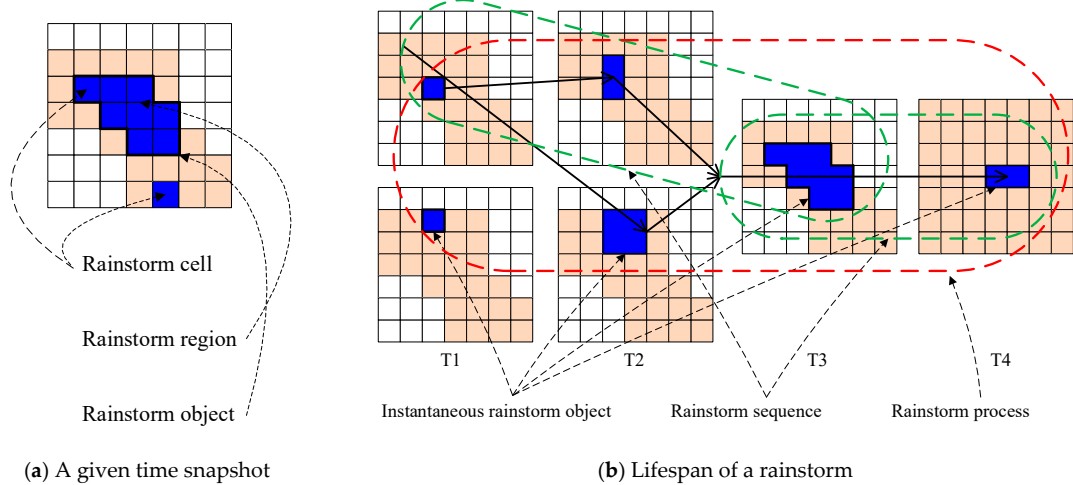

(**a**) A given time snapshot　　　　　　　　　　　　　　(**b**) Lifespan of a rainstorm

**Figure 1.** Basic concepts of a rainstorm (**a**) at a given time snapshot; (**b**) during the lifespan of the rainstorm. A solid black arrow represents a change from time snapshots T–T+1.

## 3. PoAIR Algorithm

According to the rainstorm process semantics—i.e., one rainstorm includes one or more rainstorm sequences, and one sequence includes one or more instantaneous objects—PoAIR performs three stages to obtain a rainstorm process from a time-series of raster rainfall datasets, which is shown in Figure 2. The first stage identifies instantaneous rainstorm objects in each snapshot, the second stage generates a rainstorm sequence by matching and tracking the same rainstorm objects in successive snapshots, and the third stage reconstructs a rainstorm process by linking the rainstorm sequences which overlap in spatiotemporal neighborhoods.

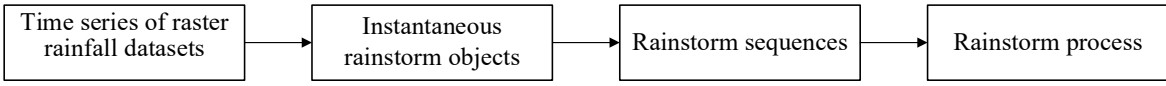

**Figure 2.** The workflow of the process-oriented algorithm for identifying and tracking rainstorms (PoAIR).

### 3.1. Identifying Instantaneous Rainstorm Objects

A rainstorm is defined when the accumulated rainfall within a specified time period exceeds a threshold, e.g., 50 mm/24 h [19]. As a time interval of obtaining rainfall datasets lower than the specified duration, PoAIR firstly determines which grid cells in a temporal domain are rainstorm cells.

According to Tobler's first law of geography, rainstorm cells tend to belong to the same group as the rainstorm cells in their neighborhood. Therefore, the PoAIR algorithm groups spatially connected rainstorm cells into rainstorm regions. Finally, the rainstorm regions are transformed into rainstorm objects, which are used to match and track rainstorm processes.

According to the definition of a rainstorm [19], the criteria for identifying rainstorm cells are as follows:

Step 1: Search for the local maximum rainfall in successive snapshots, and sort these values in descending order;

Step 2: Take the first local maximum rainfall as a searching center, and 24 h as the maximum searching time range. Then, find the successive snapshots with rainfall, which are defined as a valid time span;

Step 3: Calculate the accumulated rainfall within the valid time span. If the accumulated rainfall is over 50 mm, go to Step 4, otherwise, go to Step 6;

Step 4: Expand the valid time span forward/backward with successive snapshots until no rainfall is found, and define the expanded time span as a rainstorm duration;

Step 5: All the grid cells within the rainstorm duration are labeled as rainstorm cells;

Step 6: Go to the next local maximum rainfall value, then repeat Steps 2–5 until all the local maximum values have been analyzed.

To group grid cells into a rainstorm region, PoAIR uses spatial proximities to connect rainstorm cells. A k-neighborhood is the general spatial template used in geoinformatics for proximity analysis [24]. Using the k-neighborhood, there are three types of connections:

- Connection between rainstorm cells: If one rainstorm cell is a k-neighborhood of the other one, the former connects with the latter, otherwise, the former is regarded as noise.
- Connection between a rainstorm cell and a rainstorm region: If one rainstorm cell is a k-neighborhood of a rainstorm region, the rainstorm cell connects with the rainstorm region, otherwise, the rainstorm cell is noise.
- Connection between rainstorm regions: If at least one cell of the former rainstorm region is a k-neighborhood of the latter rainstorm region, the former connects with the latter.

Figure 3 takes the 8-neighborhood as an example to show these three types of linkage. In Figure 3c, the independent grid cells P2 and P3 are linked together, grid cells P1 and P4 are regarded as noise, grid cell P5 links with R1, rainstorm region R3 links with R4, and rainstorm region grid cell R2 links with no rainstorm grid cells or rainstorm regions.

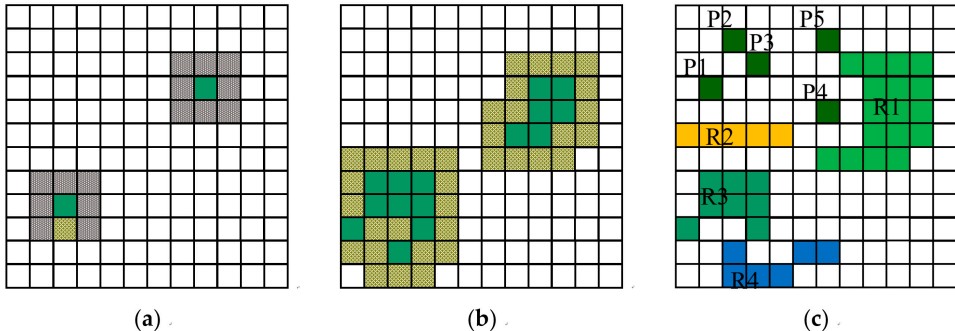

**Figure 3.** Three types of linkages with 8-neighborhood. (**a**) 8-neighborhoods of two rainstorm grid cells, (**b**) 8-neighborhoods of two rainstorm regions, and (**c**) the five independent grid cells and four spatial regions.

*3.2. Matching and Tracking Successive Rainstorm Objects in a Rainstorm Sequence*

To match instantaneous objects in successive snapshots and group them into rainstorm sequences, three steps are followed. In the first step, suitable features are selected to calculate the matching

similarities, in the second step, a matching algorithm is designed to determine which of the instantaneous objects in the current snapshot are matched with the objects in the next snapshot, and in the third step, a loop of the matching algorithm is performed until all the rainstorm objects at time snapshots have been analyzed.

For the first step, according to the definition of a rainstorm sequence, all the rainstorm objects belonging to the same sequence have similar spatial structures and thematic features, that is, during their movements, the rainstorm objects do not split or merge. Thus, the matching and tracking algorithm should impose the conditions of invariance of rotation and scale on the rainstorm objects [25]. To match and track a rainstorm sequence, this paper selects spatial features, including eccentricity ($E$), rectangularity ($Rt$), soundness ($Sd$), and a shape index ($SI$), a temporal feature, i.e., the time of occurrence, and thematic features, i.e., the mean rainfall intensity ($Rm$). The details and formulas of these features are listed in Table 1.

**Table 1.** Features selected to track rainstorm objects.

| | Name | Semantics |
|---|---|---|
| Spatial features | Centroid ($C_x,C_y$) | The position of a rainstorm, including X and Y coordinates |
| | Area ($A$) | The spatial coverage of a rainstorm object |
| | Perimeter ($P$) | The perimeter of a rainstorm object |
| | Length of Major Axis ($LoM$) | The length of the minimum bounding rectangle of a rainstorm object |
| | Width of Major Axis ($WoM$) | The width of the minimum bounding rectangle of a rainstorm object |
| | Eccentricity ($E$) | The ratio between the $WoM$ and the $LoM$ |
| | Rectangularity ($Rt$) | The ratio between $A$ and the area of the minimum bounding rectangle of a rainstorm object |
| | Roundness ($Sd$) | The ratio between the radius of the internally and externally tangent circles of a rainstorm object |
| | Shape Index ($SI$) | The ratio between the $A$ and the $P$ of a rainstorm object |
| Temporal feature | Occurrence Time | The starting time of a rainstorm object |
| Thematic features | Total precipitation | The total rainfall (in mm) of a rainstorm object in a snapshot |
| | Mean intensity | The mean rainfall intensity (in mm/h) of a rainstorm object |

As the matching objects in the successive snapshots have the minimum differences in spatial and thematic features, PoAIR uses Equations (1)–(4) to calculate the differences between rainstorm objects and build their cost function. Then, the Hungarian method is used to take the minimum difference as an object function to match and track the rainstorm objects in two successive snapshots. The Hungarian method is an optimal way to solve the assignment problem [8].

$$F = w_1 * Sd + w_2 * Td \tag{1}$$

$$Sd = w_{11} * d_E + w_{12} * d_{Rt} + w_{13} * d_{Sd} + w_{14} * d_{SI} \tag{2}$$

$$Td = \left| Rm_{t+1} - Rm_t \right| \tag{3}$$

$$\begin{cases} d_E = \left| E_{t+1} - E_t \right| \\ d_{Rt} = \left| Rt_{t+1} - Rt_t \right| \\ d_{Sd} = \left| Sd_{t+1} - Sd_t \right| \\ d_{SI} = \left| SI_{t+1} - SI_t \right| \end{cases} \tag{4}$$

where $w_1$ and $w_2$ are the weights of the spatial and thematic features and $w_{11}$–$w_{14}$ are the weights of the spatial features. As all the features are of equal importance in the implementation, $w_1$ and $w_2$ are equal to 0.5 and $w_{11}$–$w_{14}$ are equal to 0.25.

When the third step is finished, all the instantaneous rainstorm objects are grouped into either rainstorm sequences or independent rainstorm objects. Independent rainstorm objects are ones which have not been matched to another object in previous or subsequent snapshots.

### 3.3. Reconstructing a Rainstorm Process by Linking Rainstorm Sequences

Independent rainstorm objects and rainstorm sequences likely belong to the same rainstorm process if they overlap with each other [5,8,12]. Thus, we use the spatiotemporal topology between independent rainstorm objects, or between rainstorm sequences, or between an independent rainstorm object and a rainstorm sequence, as a precondition to link them into a new rainstorm sequence. Some of the new rainstorm sequences overlap with each other, which means that one splits into two or more rainstorm sequences, or two or more rainstorm sequences merge into one new sequence. That is, the rainstorm sequences which overlap with each other in space and time are composed of rainstorm processes. Then, PoAIR uses their intersections of rainstorm sequences to reshape and construct rainstorm processes. Figure 4 shows the workflow of the rainstorm reconstruction processes.

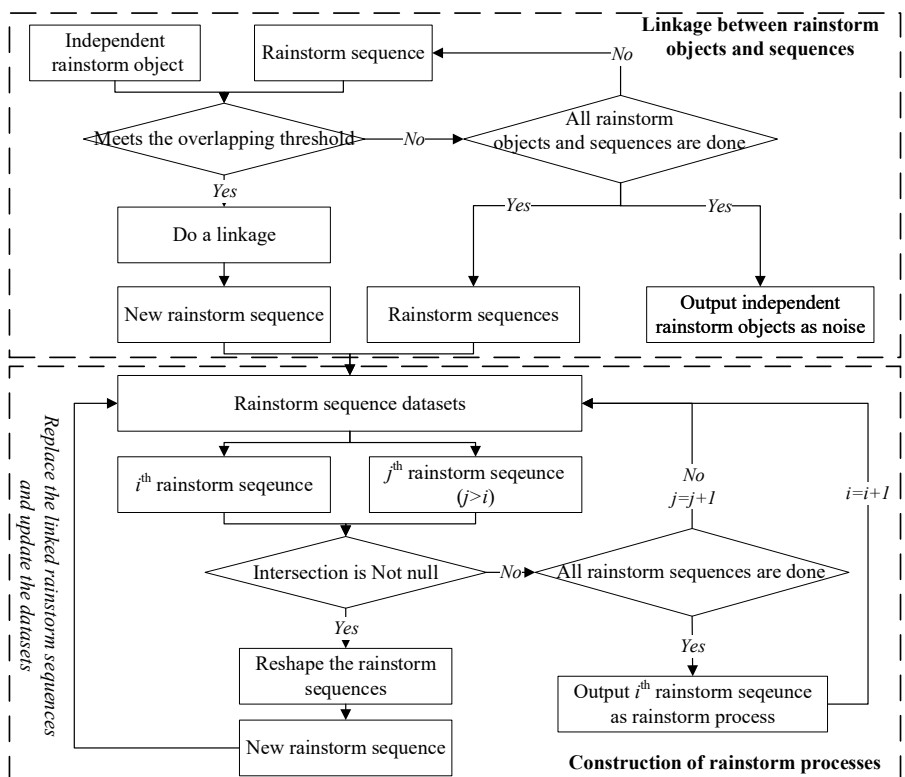

**Figure 4.** The workflow of the rainstorm reconstruction processes.

### 3.3.1. The Linkage Between Rainstorm Objects and Rainstorm Sequences

In the temporal domain, an independent rainstorm object is instantaneous, while a rainstorm sequence lasts for a specified time. These differences make the linkage strategies of rainstorm objects and rainstorm sequences different. For rainstorm objects, this paper uses an averaged moving distance, i.e., $D_{\mathrm{mean}}$, as a criterion to identify the rainstorm objects from previous and subsequent snapshots. If no object is found, the independent object is regarded as noise, or is linked the found objects. The found objects may be independent objects or objects belonging to a rainstorm sequence. For rainstorm sequences, this paper uses spatiotemporal topological relationships to find the candidates to be linked. If a rainstorm sequence overlaps in space and intersects with or is adjacent in time to a given rainstorm sequence, the latter is linked with the former; otherwise, the latter is regarded as a rainstorm process.

There are three types of linkages between rainstorm objects and rainstorm sequences:

- The linkage between rainstorm objects: Firstly, obtain the given rainstorm object and its candidates, according to the occurrence time in order, define the previous object, *preO*, and the next object, *postO*. As no motions can be obtained from the independent rainstorm objects, the PoAIR

algorithm then uses the centroid distance of the objects as a criterion to determine the presence or otherwise of linkage. If their centroid distance meets in Equation (5), a linkage is performed, and is represented in the form of *<preO->postO>*.

$$d(preO\ postO) < D_{\mathrm{mean}} \tag{5}$$

where $\mathrm{d}(preO, postO)$ means the Euclidean distance between *preO* and *postO*, $D_{\mathrm{mean}}$ is multiplied by the time interval and 58 km/h, the mean speed of typical convective rainstorm motion [26].

- The linkage between a rainstorm object and a rainstorm sequence: Firstly, find which object within a rainstorm sequence is to be linked with the given object. Then, the same linkage as between rainstorm objects is performed. This linkage is represented as:

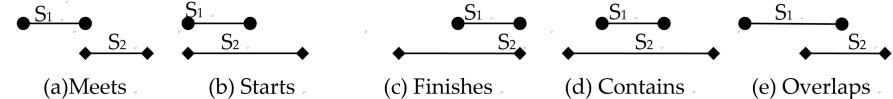

, where *S* is a rainstorm sequence and $O_S^t$ is an object occurring at time *t* within the rainstorm sequence *S*.

- The linkage between rainstorm sequences: Obtain a temporal relationship between a given rainstorm sequence and its candidate. As shown in Figure 5, there are five temporal relationships, i.e., meets, overlaps, during, starts, and finishes. The different temporal relationships require different linkage strategies.

**Figure 5.** Temporal relationships between rainstorm sequences. $S_1$: Sequence 1; $S_2$: Sequence 2.

The linkage strategy for dealing with the *Meets, Starts, and Finishes* relationships involves the following steps:

Step 1: According to the start time and end time of rainstorm sequences, identify a previous sequence and a following one, denoted as *preS* and *postS.* The previous sequence has an earlier starting time when dealing with the *Meets* and *Finishes* relationships, and an earlier ending time for the *Starts* relationship;

Step 2: Extrapolate the spatial location of *preS* in the next snapshot, denoted as *preO*. The *Meets* and *Starts* relationships use Equations (6) and (7), while the *Finishes* relationship uses Equations (9) and (10).

$$O_S^{t_e+1} = \frac{\sum_{i=t_s}^{t_e} A_i * \frac{1}{t_e-i+1} * O_S^i}{\sum_{i=t_s}^{t_e} A_i * \frac{1}{t_e-i+1}} \tag{6}$$

$$O_d^{t_e+1} = \frac{\sum_{i=t_s}^{t_e} A_i * \frac{1}{t_e-i+1} * O_d^i}{\sum_{i=t_s}^{t_e} A_i * \frac{1}{t_e-i+1}} \tag{7}$$

$$O_S^{t_s-1} = \frac{\sum_{i=t_s}^{t_e} A_i * \frac{1}{i-t_s+1} * O_S^i}{\sum_{i=t_s}^{t_e} A_i * \frac{1}{i-t_s+1}}. \tag{8}$$

$$O_d^{t_s-1} = \frac{\sum_{i=t_s}^{t_e} A_i * \frac{1}{i-t_s+1} * O_d^i}{\sum_{i=t_s}^{t_e} A_i * \frac{1}{i-t_s+1}} \tag{9}$$

where $O_S^i$, $O_d^i$, and $A_i$ are respectively the speed, direction, and area of the rainstorm object in snapshot i, $t_s$ and $t_e$ are respectively the first snapshot and the last snapshot of a rainstorm sequence, $O_S^{t_e+1}$ and $O_d^{t_e+1}$ are respectively the predicted speed and direction of a rainstorm sequence in snapshot $t_e + 1$,

$O_S^{t_s-1}$ and $O_d^{t_s-1}$ are respectively the backward speed and direction of a rainstorm sequence in snapshot $t_s - 1$;

　　　Step 3: Obtain the candidate to be linked in the *postS*, denoted as *postO*, and use Equation (8) to calculate an area overlapped degree *(AOD)* between *postO* and *preO*. If the *AOD* is above a certain threshold, a linkage is performed.

$$AOD = \frac{A(\text{Overlap})}{A(preO)} + \frac{A(\text{Overlap})}{A(postO)} \tag{10}$$

where $A(\text{Overlap})$, $A(preO)$ and $A(postO)$ are respectively the overlapped area and the area of *preO* and *postO*.

　　　*Contains* and *Overlaps* relationships perform linkages twice. The first linkage uses the starting time of rainstorm sequences to identify the *preS* and *postS*, then the same linkage as in the *Finishes* relationship is performed. The second linkage uses the end time to identify the *preS* and *postS,* then the same linkage as in the *Starts* relationship is performed.

　　　The expressions of the linkages between two rainstorm sequences are shown in Figure 6.

$O_{preS}^{0} \rightarrow \cdots \cdots \rightarrow O_{preS}^{preT} \rightarrow O_{postS}^{preT+1} \rightarrow \cdots \cdots \rightarrow O_{postS}^{postT}$

(a) *Meets* relationship

$O_{preS}^{0} \rightarrow \cdots \cdots \rightarrow O_{preS}^{t-1} \rightarrow O_{preS}^{t} \rightarrow O_{preS}^{t+1} \rightarrow \cdots \cdots \rightarrow O_{preS}^{preT}$

$O_{postS}^{0} \rightarrow \cdots \cdots \rightarrow O_{postS}^{t-1} \rightarrow O_{postS}^{t} \rightarrow O_{postS}^{t+1} \rightarrow \cdots \cdots \rightarrow O_{postS}^{preT} \rightarrow O_{postS}^{preT+1} \rightarrow \cdots \cdots \rightarrow O_{postS}^{postT}$

(b) *Starts* relationship

$O_{S2}^{t} \rightarrow O_{S2}^{t+1} \rightarrow \cdots \cdots \rightarrow O_{S2}^{T2}$

$O_{S1}^{0} \rightarrow \cdots \cdots \rightarrow O_{S1}^{t-1} \rightarrow O_{S1}^{t} \rightarrow O_{S1}^{t+1} \rightarrow \cdots \cdots \rightarrow O_{S1}^{T2} \rightarrow O_{S1}^{T2+1} \rightarrow \cdots \cdots \rightarrow O_{S1}^{T1}$

(c) *ContainS* and *Overlaps* relationships

$O_{postS}^{t} \rightarrow O_{postS}^{t+1} \rightarrow \cdots \cdots \rightarrow O_{postS}^{postT}$

$O_{preS}^{0} \rightarrow \cdots \cdots \rightarrow O_{preS}^{t-1} \rightarrow O_{preS}^{t} \rightarrow O_{preS}^{t+1} \rightarrow \cdots \cdots \rightarrow O_{preS}^{preT}$

(d) *Finishes* relationship

**Figure 6.** Expressions of linkages between rainstorm sequences. $O_S^t$ represents an object occurring at time $t$ within a rainstorm sequence $S$, i.e., *preS*, *postS*, *S1*, or *S2*.

### 3.3.2. Reconstructing Rainstorm Processes

　　　As rainstorm sequences overlapping in space and time belong to the same rainstorm process, this stage uses their intersection to reshape and construct the rainstorm processes. Additionally, during the above linkages, one rainstorm sequence may be linked with two or more rainstorm objects or sequences, that is, the generated rainstorm sequence intersects with two and more others. Thus, PoAIR designs a recursive loop to link all the related sequences to reconstruct a new rainstorm sequence, until the new sequence has no changes. The final rainstorm sequence is a rainstorm process. The recursive algorithm is as follows:

　　　Step 1: Extract the $i^{th}$ sequence from the rainstorm sequence datasets, denoted as $i^{th}$ss;

　　　Step 2: Extract the $j^{th}$ sequence, $j^{th}$ss, from the rainstorm sequence datasets, where $j$ is greater than $i$;

　　　Step 3: Calculate an intersection between the $i^{th}$ss and $j^{th}$ss. If the intersection is null, $j = j + 1$, and go to Step 2; otherwise, obtain a union of the $i^{th}$ss and $j^{th}$ss, and use the union to replace the $i^{th}$ss, update the rainstorm sequences, and go to Step 1;

　　　Step 4: $i = i + 1$, repeat Steps 1–3 until all the rainstorm sequences have no changes.

*Example 1:* There are four rainstorm sequences, i.e., *ss1*, *ss2*, *ss3*, and *ss4*. Since *ss1* and *ss2* intersect, *ss1* and *ss2* are linked into a new rainstorm sequence which replaces *ss1*, and datasets are updated. Similarly, *ss1* links *ss4* into a new sequence. After this is done two times, the datasets are updated, and *ss1* and *ss3* have no changes and are independent of each other. Thus, *ss1* and *ss3* are output as rainstorm processes. Figure 7 gives the rainstorm reconstruction process of the example.

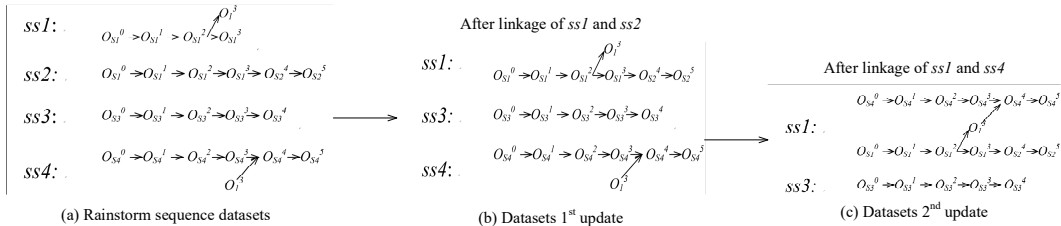

**Figure 7.** Example of a rainstorm reconstruction process.

## 4. Results and Discussion

In this section, two cases are used to evaluate our proposed PoAIR algorithm. In one, a process-oriented rainstorm dataset is simulated to investigate the performance of PoAIR compared to that of the original TITAN algorithm [8]. In the other, a real precipitation dataset is used to analyze the dynamic characteristics of a given rainstorm process.

### 4.1. Simulated Dataset

Considering the moving speed, direction, and geometric shape of the rainstorm at each time snapshot, we use the ArcGIS tool to simulate the datasets. The simulated data span 15 time steps and include 10 rainstorm processes, i.e., the simulated data consist of 170 rainstorm objects and 30 noise objects (15% of the total number of simulated objects). Among the 10 processes, rainstorm process 1, process 5, and process 9 are the basic rainstorm processes, with no merging, splitting, or merge-splitting objects, and the others are complicated processes, including at least one merging, one splitting, or one merge-splitting object. Each of the simulated processes has at least one object that moves too fast. The geometric shapes and changes over time of rainstorm processes are shown in Figure 8, and the details of the processes are shown in Table 2.

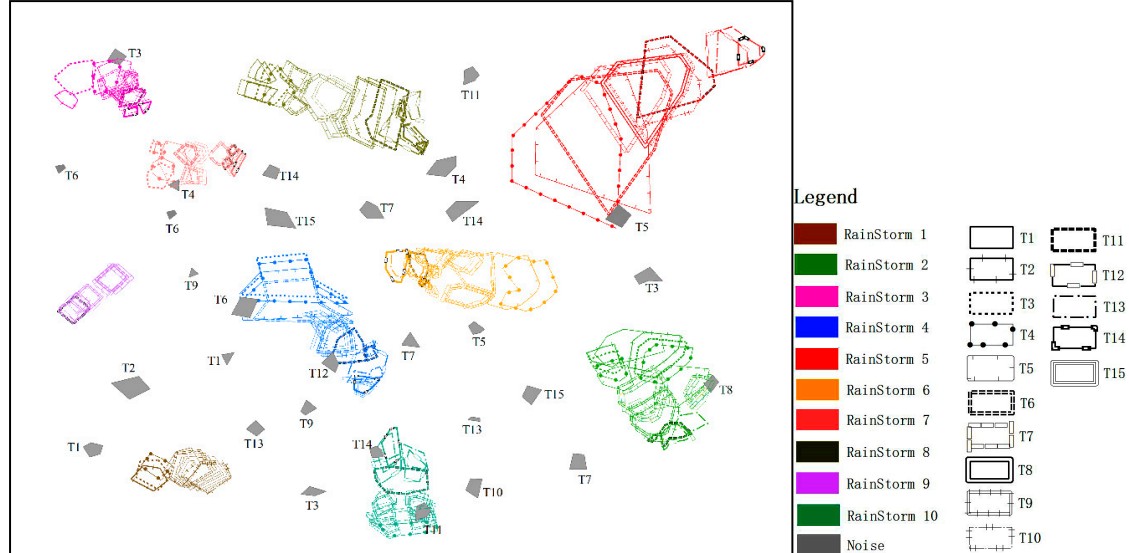

**Figure 8.** A simulated rainstorm dataset. The colors represent the different rainstorm processes, and the types of lines represent different time snapshots, i.e., T1–T15.

**Table 2.** Detailed information of the simulated rainstorm dataset shown in Figure 8.

| Name | Start–End Time | Number of Objects | Detailed Information |
|---|---|---|---|
| Rainstorm 1 | T2–T10 | 9 | A basic process has no merging, splitting, or merging-splitting objects, and turns dramatically at T4. |
| Rainstorm 2 | T1–T13 | 22 | A complicated process has three splitting objects and one merging object. |
| Rainstorm 3 | T2–T14 | 19 | A complicated process has two splitting objects, one merging object, and one merging-splitting object. |
| Rainstorm 4 | T3–T15 | 17 | A complicated process has one splitting object, two merging objects, and one merging-splitting object. |
| Rainstorm 5 | T4–T14 | 11 | A basic process has no merging, splitting, or merging-splitting objects. |
| Rainstorm 6 | T4–T14 | 22 | A complicated process has one splitting object, one merging object, and one merging-splitting object. |
| Rainstorm 7 | T3–T14 | 19 | A complicated process has one splitting object, one merging object, and one merging-splitting object. |
| Rainstorm 8 | T1–T14 | 25 | A complicated process has two splitting objects, two merging objects, and one merging-splitting object. |
| Rainstorm 9 | T5–T12 | 8 | A basic process has no merging, splitting, or merging-splitting objects, and moves with a speed of 65 km/h at T9 (exceeding the speed threshold). |
| Rainstorm 10 | T4–T14 | 18 | A complicated process has one splitting object and two merging-splitting objects. |

## 4.2. Comparisons of PoAIR and TITAN

The results of the PoAIR and TITAN algorithms are shown in Figure 9. PoAIR combines spatial similarity and thematic evolution to track rainstorm objects at successive time snapshots, which resolves the problem of excessively fast rainstorm movement. Additionally, PoAIR uses the evolution sequence, i.e., all the objects from their generation to T, to extrapolate the spatial location of the object at the time snapshot T+1, instead of only at snapshot T, which is used by TITAN. Thus, PoAIR tracked all the simulated rainstorm processes except process 1, process 6, and process 9 (Figure 9a). As it uses the overlapped-area threshold to track an object at successive time snapshots, TITAN has great difficulty tracking rainstorm processes, and identifies each of the simulated rainstorm processes as two or more independent processes (Figure 9b). The main reason for this is that each simulated rainstorm process has at least one object that moves too fast.

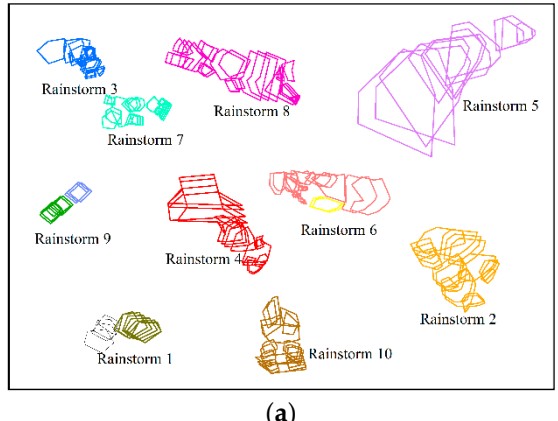 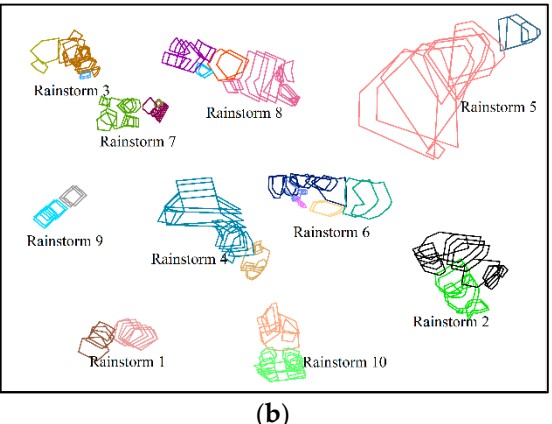

(**a**)    (**b**)

**Figure 9.** Results of the PoAIR and TITAN algorithms. (**a**) Results of the PoAIR algorithm using simulated data. (**b**) Results of the TITAN algorithm using simulated data.

Compared with TITAN, PoAIR has an improved capability to track fast-moving rainstorms. However, when a rainstorm reaches its maximum threshold of movement speed, i.e., 58 km/h in this

paper, PoAIR has no ability to track it. For example, PoAIR identified Rainstorm 9 as two independent processes since at T9 the rainstorm moved with a speed of 65 km/h. Additionally, PoAIR also has difficulty tracking rainstorms which dramatically change their movement direction. The reason for this is that the spatial location of the rainstorm at time T+1 is extrapolated from its generation to time T, and when it changes direction, the distance between the rainstorm's actual location and the extrapolated one exceeds the maximum search radius of PoAIR. For example, Rainstorm 1 turns dramatically to the right at T4, which leads to the position of the rainstorm at T5 exceeding the maximum search radius of PoAIR, and thus Rainstorm 1 was identified as two independent processes.

As PoAIR uses a process-sequence-state semantic to reconstruct rainstorm processes, it is more insensitive to noise than the TITAN algorithm (Figure 9b). In this paper, noise is defined as an independent object, which is neither connected with other objects in space nor evolves in successive time snapshots.

To assess the overall performance of PoAIR compared with the original TITAN algorithm, the probability of detection (*POD*), false alarm ratio (*FAR*), and critical success index (*CSI*) scores, which are well-known indicators in precipitation evaluations [5,27], are computed by following Equation (11). Table 3 shows their comparisons in a view of rainstorm processes, and Table 4 shows their comparisons in a view of rainstorm objects. In Table 3, if a rainstorm is identified as two or more rainstorm processes, the rainstorm processes, including the more objects, are used to calculate the evaluation indicators.

$$
\begin{cases}
POD = \frac{n_{hits}}{n_{hits}+n_{misses}} \\
FAR = \frac{n_{false}}{n_{hits}+n_{false}} \\
CSI = \frac{n_{hits}}{n_{hits}+n_{misses}+n_{false}}
\end{cases}
\tag{11}
$$

where $n_{hits}$ occurs when both the algorithm and the human observation capture a rainstorm object; $n_{misses}$ occurs when human observation identifies a rainstorm object that the algorithm does not; and $n_{false}$ occurs when the algorithm considers a false rainstorm object.

**Table 3.** A comparison of the rainstorm processes identified by the process-oriented algorithm for identifying and tracking rainstorms (PoAIR) and the Thunderstorm Identification, Tracking, Analysis and Nowcasting (TITAN) algorithm.

| | Rainstorm 1 | | Rainstorm 2 | | Rainstorm 3 | | Rainstorm 4 | | Rainstorm 5 | |
|---|---|---|---|---|---|---|---|---|---|---|
| | **PoAIR** | **TITAN** | **PoAIR** | **TITAN** | **PoAIR** | **TITAN** | **PoAIR** | **TITAN** | **PoAIR** | **TITAN** |
| *POD* (%) | 60.00 | 60.00 | 90.91 | 54.54 | 94.70 | 78.95 | 100.00 | 58.82 | 100.00 | 72.72 |
| *FAR* (%) | 0.00 | 0.00 | 0.00 | 7.69 | 0.00 | 6.25 | 0.00 | 16.67 | 0.00 | 11.11 |
| *CSI* (%) | 60.00 | 60.00 | 90.91 | 52.17 | 94.70 | 75.00 | 100.00 | 52.63 | 100.00 | 66.67 |
| | **Rainstorm 6** | | **Rainstorm 7** | | **Rainstorm 8** | | **Rainstorm 9** | | **Rainstorm 10** | |
| | **PoAIR** | **TITAN** | **PoAIR** | **TITAN** | **PoAIR** | **TITAN** | **PoAIR** | **TITAN** | **PoAIR** | **TITAN** |
| *POD* (%) | 90.91 | 45.00 | 100.00 | 57.89 | 100.00 | 48.00 | 62.50 | 62.50 | 100.00 | 66.67 |
| *FAR* (%) | 0.00 | 0.00 | 0.00 | 0.00 | 0.00 | 0.00 | 0.00 | 0.00 | 0.00 | 14.28 |
| *CSI* (%) | 90.91 | 45.00 | 100.00 | 57.89 | 100.00 | 48.00 | 62.50 | 62.50 | 100.00 | 60.00 |

**Table 4.** A comparison of rainstorm statuses identified by the PoAIR and TITAN algorithms.

| | Production | | Developing | | Merging | | Splitting | | Merging-Splitting | | Termination | |
|---|---|---|---|---|---|---|---|---|---|---|---|---|
| | **PoAIR** | **TITAN** | **PoAIR** | **TITAN** | **PoAIR** | **TITAN** | **PoAIR** | **TITAN** | **PoAIR** | **TITAN** | **PoAIR** | **TITAN** |
| *POD* (%) | 93.33 | 93.33 | 91.74 | 74.31 | 88.89 | 44.44 | 72.73 | 54.54 | 85.71 | 42.86 | 84.21 | 89.47 |
| *FAR* (%) | 30.00 | 50.00 | 3.85 | 10.00 | 11.11 | 20.00 | 0.00 | 45.45 | 0.00 | 0.00 | 20.00 | 55.26 |
| *CSI* (%) | 66.67 | 48.38 | 88.50 | 68.64 | 80.00 | 40.00 | 72.73 | 37.50 | 85.71 | 42.86 | 69.56 | 42.50 |

For tracking rainstorm processes, PoAIR performs better than TITAN in terms of *POD*, *FAR*, and *CSI* for both basic and complicated rainstorms. Regarding basic rainstorms, both PoAIR and TITAN identified Rainstorm 9 and Rainstorm 1 as two independent processes, i.e., they had the same

detection abilities. For Rainstorm 5, TITAN identified it as two independent processes, while PoAIR identified it as one. Regarding complicated rainstorms, PoAIR combines spatial similarities and the evolution of rainstorm sequences to track rainstorm processes, which allows it to track fast-moving rainstorms. Thus, PoAIR has a higher *POD*, a lower *FAR*, and a higher *CSI* than TITAN.

In statuses of rainstorm production, development, and termination, both PoAIR and TITAN have high values of *POD* and a high value of *CSI*, with PoAIR having slightly higher values than TITAN. As it uses evolution sequences of rainstorms to track rainstorm processes, PoAIR mistakes an independent rainstorm object at the production or termination stage as noise, e.g., the termination stage of Rainstorm 2 has a *FAR* value of 20.00% and the production stage of Rainstorm 3 has a *FAR* value of 26.32%. TITAN identified all of the simulated rainstorms as two rainstorm processes, which led to the numbers of detected production/termination stages being almost twice that of the actual ones, and the value of FAR reaches 50.00%. For statuses of merging, splitting, and merging-splitting, PoAIR has an obvious advantage over TITAN. The former has a higher value of *POD*, a higher value of *CSI*, and a lower value of *FAR* than the latter. Although both of the algorithms use an area-overlapping threshold to link the rainstorms at successive time snapshots, PoAIR extrapolates a spatial location at the T+1 snapshot by using the spatial information of rainstorm sequences from their generation to the T snapshot, while TITAN uses only the spatial information from the T snapshot. The extrapolating algorithm with evolution sequences improves the rainstorm detection capability of the PoAIR algorithm.

### 4.3. Dynamic Characteristics of a Given Rainstorm Process

The study area is mainland China, and the real precipitation dataset comes from the Integrated Multi-satellitE Retrievals for Global Precipitation Mission (GPM/IMERG) final precipitation products, version 05B, which provides global precipitation datasets with a spatial resolution of $0.1 \times 0.1°$ and a temporal resolution of 0.5 h [28]. This dataset has been evaluated and widely used in China [29–31]. The time span of the dataset used is from March 2014 to December 2017, and the spatial range of the dataset covered in this paper is shown in Figure 10.

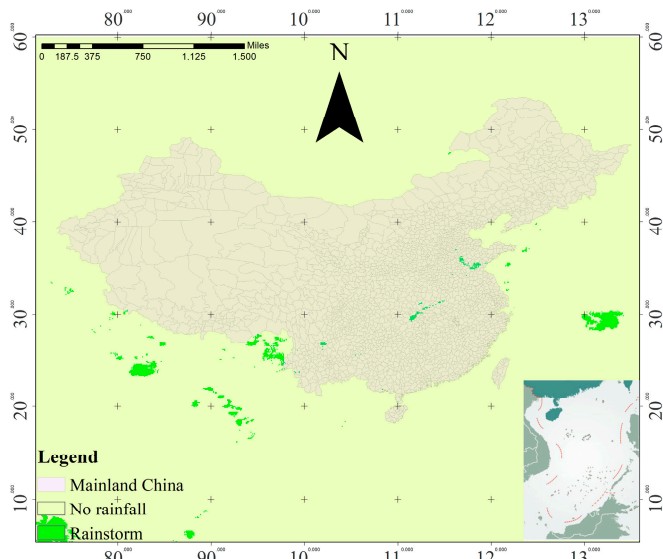

**Figure 10.** Research area and remote sensing dataset coverage.

Based on our analysis and existing knowledge, PoAIR chooses 8 neighborhoods [9,32] to transform the spatially-connected grid cells to spatial rainstorm objects, an area-overlapping threshold of 0.6 [8,12], and a maximum research radius of 58 km to link the rainstorm sequences [26]. There is a total of 300,201 rainstorm processes in the dataset. Among them, the rainstorms which occurred during the period of 18–21 July 2016, involved a complicated rainstorm process, which included 373 sequences and 2560 objects, and 254 merging, 263 splitting, and 150 merging-splitting behaviors. Figure 11

shows the rainstorm process, and mainly displays the key time snapshots when splitting, merging, or merging-splitting behaviors occur, or when local stations publish altering information.

The rainstorm processes shown in Figure 11 originated in Shaanxi, Hubei, and Hunan provinces. As time went by, the rainstorms gradually moved to Northeast China with splitting, merging, and merging-splitting behaviors, and eventually terminated in Liaoning Province. At 01:00 on July 20, rainstorm objects originated in Shaanxi Province, and those in Hunan and Hubei provinces merged in the middle of Henan Province and then moved Northeast. At 02:30 on July 20, this merged rainstorm object split into two parts, one of which continued to weaken and terminated in Henan Province at 18:00 on July 20; the other part kept on moving Northeast and then split into two parts at 00:30 on July 20. Meanwhile, other rainstorm objects affecting Beijing, Tianjin, and Hebei provinces continued to strengthen and moved towards Liaoning Province. At 09:00 on July 21, this rainstorm process split again, and each part continued to weaken. Finally, it terminated in Liaoning Province at 19:30 on July 21. As can also be seen in Figure 11, this rainstorm process closely coincides in space and time with the rainstorm warning information issued by multiple meteorological stations.

Figure 12 shows one of the basic rainstorm sequences which occurred between 22:00 on July 20 –01:00 on July 21, including seven rainstorm objects. The sequence only had developing behaviors, and its rainstorm objects had a similar movement direction and similar geometrical and thematic features, as shown in Table 5. This similarity proves that the PoAIR algorithm produces desired results.

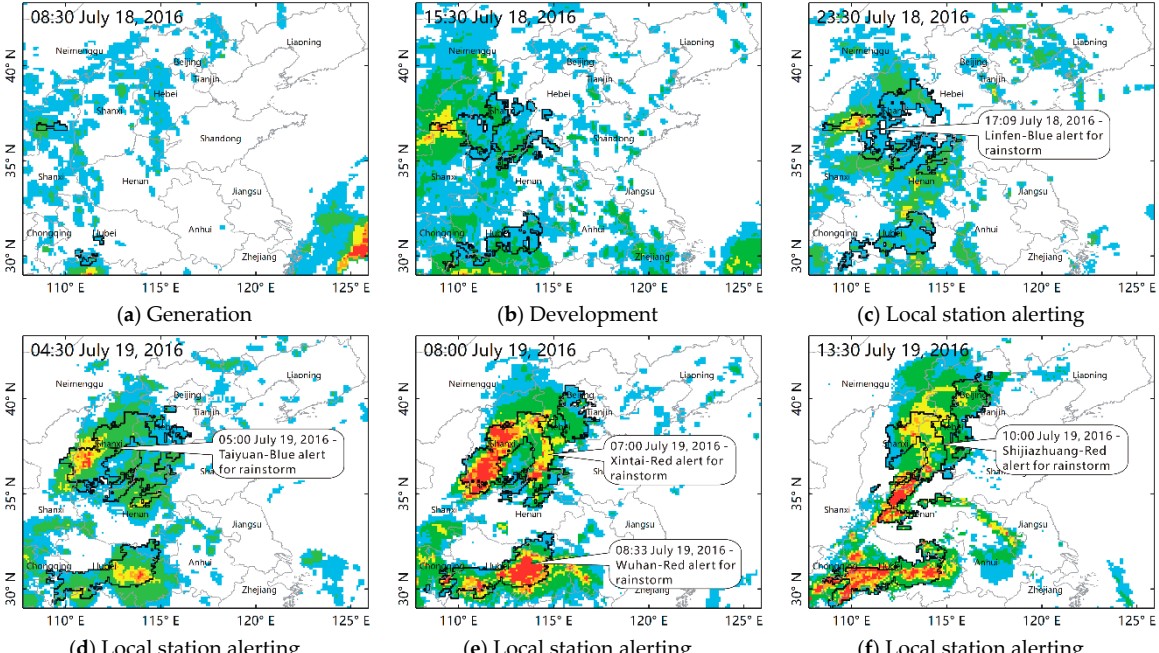

**Figure 11.** *Cont.*

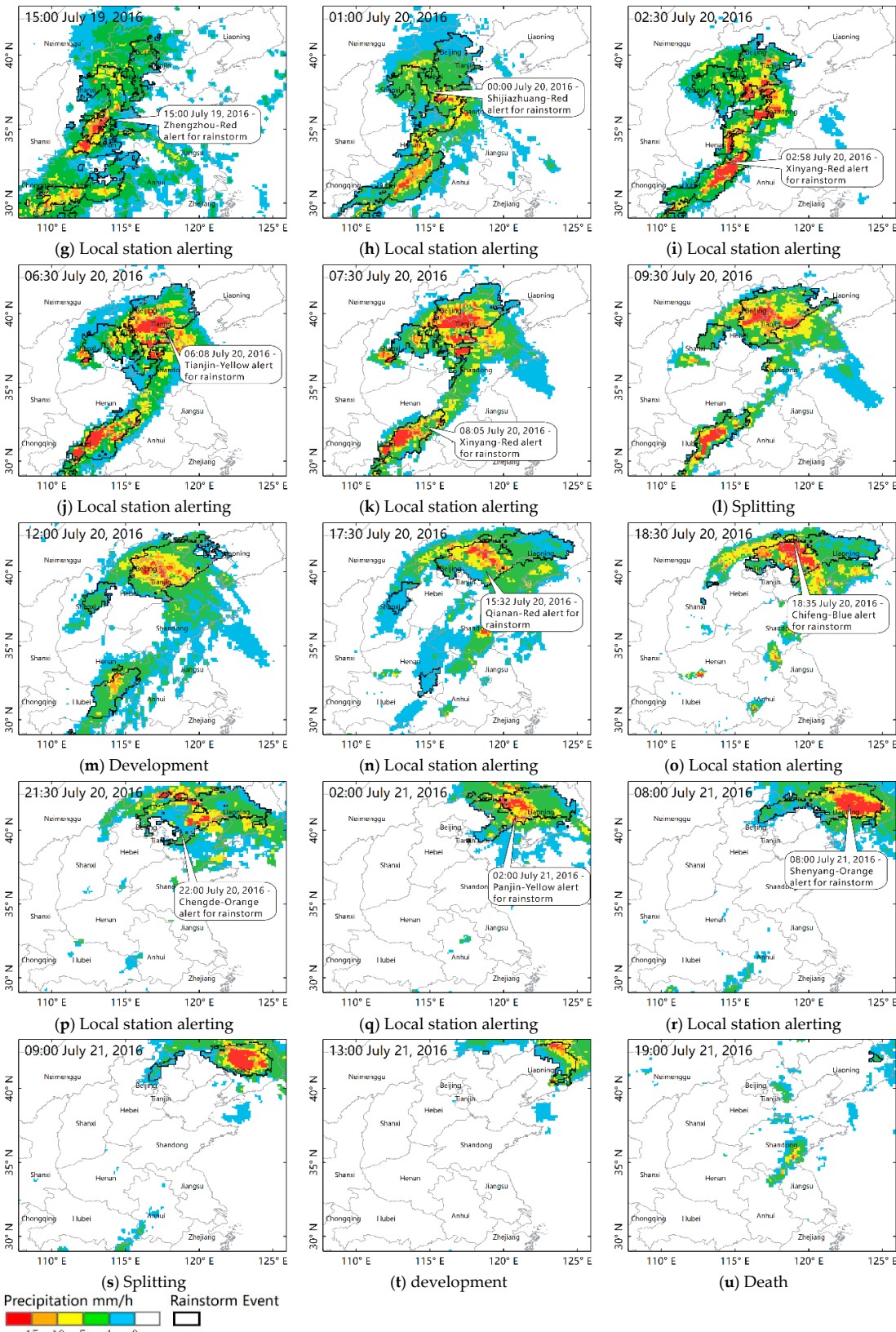

**Figure 11.** The dynamic characteristics of rainstorm processes over mainland China between 18–21 July 2016. The background is rainfall from GPM/IMERG products, the black lines denote rainstorm objects, and the text balloons present spatial and temporal information of rainstorm warnings from stations.

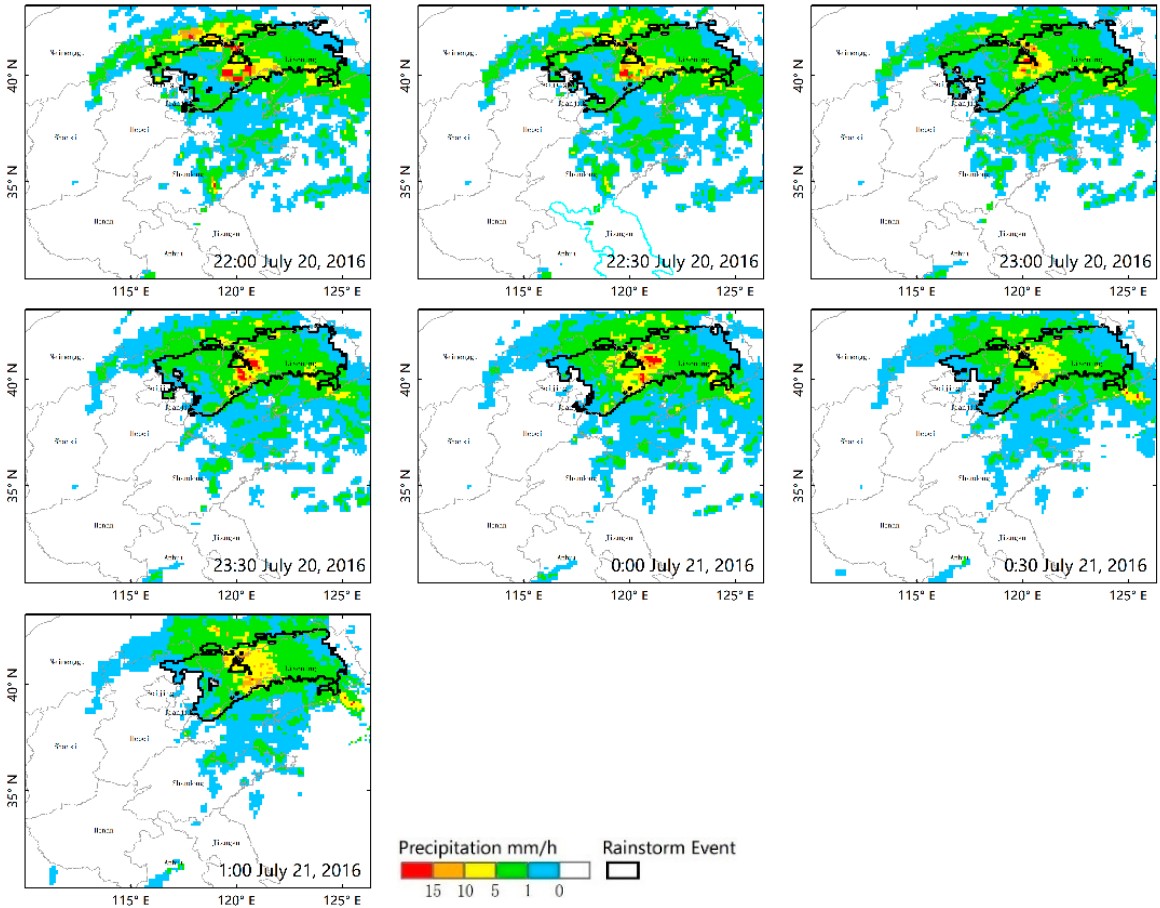

**Figure 12.** The dynamic characteristics of a rainstorm sequence which occurred between 22:00 on July 20 and 01:00 on July 21.

**Table 5.** The geometrical and thematic features of the rainstorm sequence shown in Figure 12.

| Rainstorm Object | Geometrical Features | | | | Thematic Feature |
|---|---|---|---|---|---|
| | Eccentricity (*E*) | Rectangularity (*Rt*) | Roundness (*Sd*) | Shape Index (*SI*) | Mean Intensity |
| 22:00, July 20 | 0.391 | 0.564 | 0.262 | 0.359 | 1.597 |
| 22:30, July 20 | 0.395 | 0.556 | 0.266 | 0.356 | 1.483 |
| 23:00, July 20 | 0.408 | 0.600 | 0.248 | 0.364 | 1.345 |
| 23:30, July 20 | 0.406 | 0.598 | 0.277 | 0.393 | 1.476 |
| 00:00, July 21 | 0.407 | 0.566 | 0.259 | 0.379 | 1.609 |
| 00:30, July 21 | 0.404 | 0.554 | 0.232 | 0.378 | 1.432 |
| 01:00, July 21 | 0.423 | 0.553 | 0.250 | 0.385 | 1.642 |
| Mean | 0.405 | 0.570 | 0.256 | 0.373 | 1.512 |
| Standard deviation | 0.010 | 0.019 | 0.013 | 0.013 | 0.100 |

## 5. Conclusions

A new methodology for identifying and tracking rainstorms, called PoAIR, has been presented using long-term raster datasets. PoAIR overcomes two main issues of existing tracking algorithms: (1) It matches rainstorm objects that move at high speed across successive snapshots, and (2) it resolves deficiencies of splitting and merging. For the first issue, PoAIR combines geometric similarities and thematic evolutions as a cost function to track rainstorm objects across successive snapshots; for the second, PoAIR extrapolates the spatial locations of rainstorm objects at snapshot T+1 using the rainstorm sequence from its generation to snapshot T, rather than using the rainstorm object only at snapshot T as the TITAN algorithm does.

Thus, a comparison between the PoAIR and TITAN algorithms has shown that PoAIR performs better than TITAN in terms of *POD*, *FAR*, and *CSI* for both basic and complicated rainstorm processes,

and that PoAIR has an obvious advantage over TITAN when dealing with splitting, merging, and merging-splitting rainstorm objects. Additionally, a rainstorm outbreak on 18–21 July 2016 spanning a wide swath from Southern to Northeastern China was utilized to demonstrate the development and application of the PoAIR algorithm. PoAIR identified all the rainstorm sequences and rainstorm objects, as well as the dynamic behaviors among successive rainstorm objects, i.e., developing, merging, splitting, and splitting-merging.

This paper adopts the widely used area-overlapping threshold and maximum search radius to track rainstorm processes. To assess the robustness of PoAIR, its performance must be further evaluated with more rainstorm processes and experiments. Additionally, PoAIR has difficulty dealing with objects whose movement direction changes dramatically. For this, prior knowledge, e.g., of atmospheric wind fields or atmospheric circulation, will be used to calculate the spatial location at snapshot T+1, instead of using the extrapolating approach based on sequences.

**Author Contributions:** Conceptualization, C.X.; data curation, C.W.; funding acquisition, C.X.; methodology, C.X. and J.L.; software, G.Y. and C.W.; validation, J.L. and G.Y.; writing—original draft, C.X.; writing—review and editing, C.X.

**Funding:** This research was supported in part by the National Key Research and Development Program of China under Grant 2017YFB0503605 and Grant 2016YFA0600304; in part by the Strategic Priority Research Program of the Chinese Academy of Sciences under Grant XDA19060103; and in part by the National Natural Science Foundation of China under Grant 41671401.

**Acknowledgments:** We thank the data working group of the Global Precipitation Missions for providing the research data, i.e., GPM/IMERG final precipitation products, version 05B.

**Conflicts of Interest:** The authors declare no conflict of interest.

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
