# Peer review of "A Process-Oriented Method for Tracking Rainstorms with a Time-Series of Raster Datasets"

_applsci, doi:10.3390/app9122468_

Round 1

Reviewer 1 Report

In general, the article is readable. Figures need attention. The content in figures is not readable.

However, the article needs a final proofreading. 

At places some semantic and syntactic issues. 

The following sentences including figures need attention:

Line Number: 10 - what are "fine spatial structures"?

Line Numbers 119-120, sentence is incomplete?

Line number 168: what is "until all times"? not clear

Line Number 196: reword the sentence

Section 4.1 is empty?

Figure 8, the legend is poorly legible.

In Figure 9, the object names are not legible

Sentences 344 and 345 may be rephrased

Line numbers 370-374, the sentences are not clear

Sentences 396-398 are not clear

Figure 10 has so many panels, the information within each panel is not clear. The figure may be split, making information within each panel more clear for the audience.

Line numbers 420-421, what do you mean by "reasonable results"? A colour bar could be added in Figure 11.

In Table 5, can these attributes be plotted in a plot view, adding more rainstorm objects?

Referencing style may be checked

Author Response

Reviewer1

Comments 1: The content in figures is not readable.

Answer: Figure 8, Figure 9 and Figure 10 are redrawn, and Figure 11 adds a legend.

Comments 2: Line Number: 10 - what are "fine spatial structures"?.

Answer: We are sorry for unclear representation about the fine spatial structures of rainstorms. Here, fine spatial structures mean that the rainstorms have more precise spatial boundary and rainfall.

Comments 3: Line Numbers 119-120, sentence is incomplete?

Answer: Thanks a lot for your comments. The first paragraph of Section3 (Line 113-line 119) is rewritten.

Comments 4: Line number 168: what is "until all times"? not clear.

Answer: Thanks. This sentence has been rewritten as “ a loop of the matching algorithm is performed until all the rainstorm objects at time snapshots have been analyzed”.

Comments 5: Line Number 196: reword the sentence

Answer: Done.

Comments 6: Section 4.1 is empty?

Answer: Thank you for your invaluable suggestions. According to your suggestions, Titles of subsection of Section 4 are recognized as follows. Section 4.1 is Simulated dataset, Section 4.2 is Comparisons of PoAIR and TITAN, and Section 4.3 is Dynamic characteristics of a given rainstorm process.

Comments 7: Figure 8, the legend is poorly legible.

Answer: The legend is redrawn.

Comments 8: In Figure 9, the object names are not legible

Answer: Figure 9 is redrawn.

Comments 9: Sentences 344 and 345 may be rephrased

Answer: Done.

Comments 10: Line numbers 370-374, the sentences are not clear

Answer: We are sorry make it unclear for the reviewer and readers. The revised manuscript rewrites these sentences.

Comments 11: Sentences 396-398 are not clear

Answer: We are sorry make it unclear for the reviewer and readers. The revised manuscript rewrites these sentences.

Comments 12: Figure 10 has so many panels, the information within each panel is not clear. The figure may be split, making information within each panel more clear for the audience.

Answer: The revised manuscript reorganizes the Figure 10, and for each panel, we give its key information, i.e. altering information, or merging, splitting behavior information.

Comments 13: Line numbers 420-421, what do you mean by "reasonable results"? A colour bar could be added in Figure 11.

Answer: According to the idea of our algorithm, the rainstorm objects belonging to the same rainstorm sequence should have the similar geometrical and thematic features. And Figure 11 proves this argument. We think this is reasonable, maybe this is not suitable, thus, the revised manuscript replaces the “reasonable results” with the “desired results”.

Also, Figure 11 adds a color bar (Legend).

Comments 14: In Table 5, can these attributes be plotted in a plot view, adding more rainstorm objects?

Answer: Thanks for your suggestions. Table 5 can be plotted in a plot view, shown as the following figure. As the selected sequence only includes 7 rainstorm objects, there is no more objects be added into the Table 5 or the Figure. As only7 rainstorm objects are used, we think the Table may be more suitable than Figure. If more rainstorm objects are used, the Figure are more suitable than Table. Thus, the revised manuscript retains the Table5.

Comments 15: Referencing style may be checked.

Answer: Thanks. All the references are checked again.

Reviewer 2 Report

Congratulation to author.

The article is interesting, and the subject is worthy of research. I find it really well-thought-of, well written and finally well balanced between methodology, results and conclusions.

The proposed process based model is well described. The strongest part in my opinion is section 4 that presents performance of described methodology in relation both to other existing model and a real dataset. For more clear structure I would change the chapter name to more classic “Results and discussion”.

Just to advice in the Conclusion chapter authors could underline a bit stronger the innovation aspect of the work discussed in the paper especially they proved that this methodology gives really good results compared to other similar works. Please emphasize this and than show in general what is new in this approach.

Additionally I would advise to add one general figure in chapter 4 representing  the test area of China mainland within a bigger perspective with north direction and scale before figure 10 which presents the dynamic characteristics of rainstorm.

Author Response

Comments 1: The proposed process based model is well described. The strongest part in my opinion is section 4 that presents performance of described methodology in relation both to other existing model and a real dataset. For more clear structure I would change the chapter name to more classic “Results and discussion”. 

Answer: Done.

Comments 2: Just to advice in the Conclusion chapter authors could underline a bit stronger the innovation aspect of the work discussed in the paper especially they proved that this methodology gives really good results compared to other similar works. Please emphasize this and than show in general what is new in this approach.

Answer: Thanks for your comments. The second paragraph in Conclusion Section is to give the advantages of our algorithm compared to other similar works. The revised manuscript rewrites this paragraph.

Comments 3: Additionally I would advise to add one general figure in chapter 4 representing the test area of China mainland within a bigger perspective with north direction and scale before figure 10 which presents the dynamic characteristics of rainstorm.

Answer: Thanks for your suggestions. The revised manuscript adds one figure representing the test area of China mainland within a bigger perspective with north direction and scale.
